# A Constellation-Aware Transformer for Nonlinear Channel Equalization

## Abstract

Decoding signals over unknown channels with minimal pilot overhead is a critical challenge in communications. Existing deep learning approaches struggle to model long-range temporal dependencies. Conversely, off-the-shelf Transformers, while powerful sequence models, are domain-agnostic and inefficiently learn the channel's physical properties from scarce data. We introduce the *Constellation-Aware Transformer* (CAT), a novel architecture that integrates fundamental communication principles into the Transformer model. CAT is composed of a stack of custom *TransFIRmer* blocks, which redesign the standard Transformer to be constellation-aware. Each block facilitates deep interaction between the received signals and the ideal constellation geometry via a specialized attention mechanism. Furthermore, it replaces the standard feed-forward network with a two-stream architecture: a bidirectional Finite Impulse Response (FIR)-inspired filter processes the signal representations for robust deconvolution, while a parallel MLP refines the constellation representations. In the challenging semi-supervised setting, CAT achieves superior performance across multiple noisy channels, significantly outperforming other baselines, with using fewer pilot signals.

## 1 Introduction

Deep learning methods for communications over unknown channels have attracted considerable interest recently (O'Shea & Hoydis, 2017; Bennatan et al., 2018; Nachmani et al., 2018; O'Shea et al., 2018; Shlezinger et al., 2020). A central challenge in this domain is to minimize the amount of pilot data required for reliable decoding, as these known symbols constitute a transmission overhead that reduces the overall data rate (Shlezinger et al., 2021). While classical methods like Expectation-Maximization (EM) offer a path to unsupervised equalization (Tong & Perreau, 1998; Dempster et al., 2018), their performance can be limited. To improve upon this, deep generative models, specifically Variational Autoencoders (VAEs) (Kingma & Welling, 2014), have been successfully adapted to a semi-supervised learning (SSL) framework (Kingma et al., 2014). This approach, which uses both labeled pilots and unlabeled payload data, has demonstrated a significant reduction in the required number of pilots (Burshtein & Bery, 2023; 2024).

However, the encoders in these systems, typically implemented with multilayer perceptron (MLP) or convolutional neural networks (CNNs), may not fully capture the complex temporal dependencies present in communication signals. The remarkable success of the Transformer architecture (Vaswani et al., 2017) in sequence modeling motivates its application to this domain. A direct, off-the-shelf adaptation, however, is suboptimal (Choukroun & Wolf, 2024). A generic Transformer is a powerful but domain-agnostic model that must learn the underlying physics of the problem—such as the discrete, geometric nature of the constellation and the filtering properties of the channel—entirely from data. This is inefficient and forfeits the advantage of decades of communication theory.

We argue that the path to superior performance lies not merely in applying a more powerful architecture, but in fundamentally redesigning its internal components to incorporate known principles. Here, we draw inspiration from a highly successful paradigm in natural language processing (NLP): *early and deep interaction*. In tasks like dense retrieval and multi-document processing, state-of-the-art models have shown that jointly processing a query and a document from the very first layer, allowing for fine-grained, token-level attention throughout the network, dramatically outperforms methods that process them independently and only compare final vector representations (Humeau

et al., 2019; Fang et al., 2020; Xiao et al., 2022; Liu et al., 2023). This principle of avoiding premature summarization by enabling early interaction is directly applicable to our problem.

We introduce the *Constellation-Aware Transformer (CAT)*, an architecture composed of a stack of novel *TransFIRmer* blocks, illustrated in Figure 1. Each TransFIRmer block is a self-contained processing block that redesigns the standard Transformer encoder by integrating two key innovations motivated by this principle:

1. Constellation-Aware Attention (CAT): A custom attention mechanism that co-processes the received signals alongside the ideal constellation representations from the very first layer. This implements the principle of early interaction, providing the model with an explicit, perfect prior of the target symbol space that guides the equalization process at every stage.

2. FIR-Inspired Feed-Forward Network: The standard position-wise MLP is replaced with a specialized, two-stream feed-forward network. This component, inspired by classical Finite Impulse Response (FIR) filters, is structurally tailored for the task of channel deconvolution, allowing it to learn an inverse channel filter more effectively than a generic MLP.

We demonstrate that CAT achieves state-of-the-art performance by creating a synergy between powerful models from the discrete, symbolic world of NLP to solve the inherently continuous-valued problem of signal equalization in communication theory.

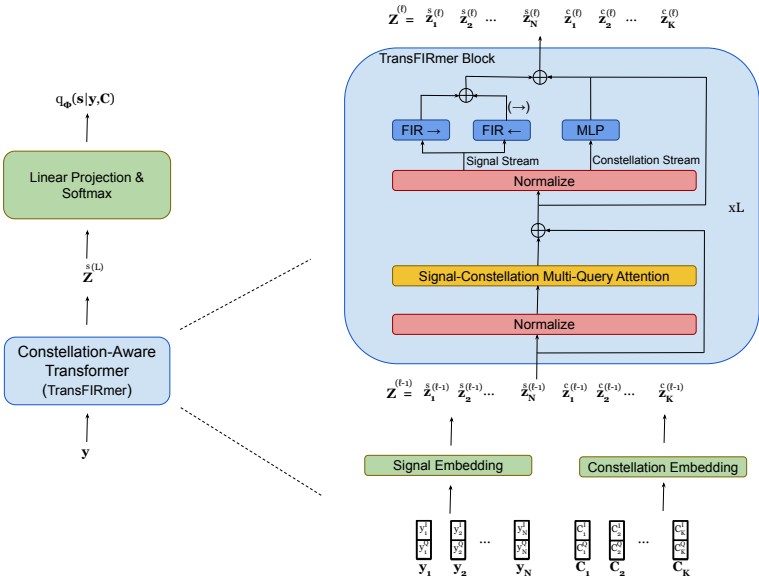

Figure 1: The architecture of our Constellation-Aware Transformer (CAT), which processes the received signals $\mathbf{y}$ and ideal constellation symbols $\mathcal{C}$, through a stack of $L$ TransFIRmer blocks.

## 2  PROBLEM FORMULATION AND SETUP

We consider a block of $N$ symbols, $(s_1, \ldots, s_N)$, transmitted over an unknown channel. Each symbol, $s_i \in \{1, \ldots, K\}$, is drawn independently uniformly. The symbol $s_i$ is mapped to a corresponding complex signal from a fixed constellation $\mathcal{C}$ of size $K$. For a more convenient processing, we represent these signals as real-valued vectors – We formally define an equivalence between a complex number in $\mathbb{C}$ and its real-vector representation in $\mathbb{R}^2$. These notations are used interchangeably for all physical quantities (e.g., transmitted and received signals, channel taps, and noise). Specifically, the ideal transmitted signal is denoted $\mathbf{x}_i = (x_i^I, x_i^Q) \in \mathbb{R}^2$, where $x_i^I$ and $x_i^Q$ are the in-phase and quadrature components. We let $\mathbf{x}(k)$ be the vector corresponding to symbol $s_i = k$.

The channel is unknown at the receiver. However, a small subset of the transmitted symbols, $\{s_i\}_{i=1}^{N_p}$, are known as pilot symbols, where $N_p \ll N$. The remaining symbols, $\{s_i\}_{i=N_p+1}^{N}$, constitute the unknown payload.

## 2.1 CHANNEL MODELS

We consider two classes of channels: memoryless channels and channels with finite memory.

### 2.1.1 MEMORYLESS CHANNELS

In a memoryless channel, the received signal at a given time instant depends only on the signal transmitted at that same instant. This process can include nonlinear distortions at the transmitter, such as I/Q imbalance or effects from components like power amplifiers. We model this entire end-to-end transformation as a complex, unknown function $h(\cdot)$. The received signal $\mathbf{y}_i$ is the result of this nonlinear function applied to the ideal signal $\mathbf{x}_i$, corrupted by additive noise $\mathbf{n}_i$, $\mathbf{y}_i = h(\mathbf{x}_i) + \mathbf{n}_i$, where $\mathbf{y}_i = (y_i^I, y_i^Q) \in \mathbb{R}^2$. The conditional probability density function (PDF) $p(\mathbf{y}_i|s_i)$ is thus the distribution induced by this memoryless process.

### 2.1.2 CHANNELS WITH FINITE MEMORY

In channels with finite memory, the received signal $\mathbf{y}_i$ is affected by intersymbol interference (ISI), meaning it depends on a sequence of past transmitted signals. This is common in wireless communications due to multipath propagation. Following the model in Burshtein & Bery (2024), this can be described by a two-stage process: first, a memoryless nonlinearity $g(\cdot)$ at the transmitter (e.g., from power amplifiers), and second, a linear filter representing the channel's impulse response. The received signal at time $i$ is a convolution of the channel's impulse response with the sequence of (potentially distorted) transmitted signals, $\mathbf{y}_i = \sum_{l=0}^{L-1} \mathbf{h}_l g(\mathbf{x}_{i-l}) + \mathbf{n}_i$, where $\{\mathbf{h}_l\}_{l=0}^{L-1}$ are the complex-valued filter taps of the channel impulse response (of length $L$), and $g(\cdot)$ is an unknown nonlinear function. Both the taps $\{\mathbf{h}_l\}$ and the function $g(\cdot)$ are unknown to the receiver.

## 2.2 DECODING AND LEARNING FRAMEWORKS

A standard supervised learning approach is to train a decoder using only the pilot data. The nature of the decoder, however, depends on the channel type.

- For *memoryless channels*, the decoder network, parameterized by $\phi$, can process each sample independently. Its input is the current sample $\mathbf{y}_i$, and it outputs a posterior probability distribution $q_\phi(s|\mathbf{y}_i)$.

- For *channels with memory*, the decoder must process the entire sequence of received signals $\mathbf{y} = (\mathbf{y}_1, \dots, \mathbf{y}_N)$ to resolve the interference and decode a single symbol $s_i$. In this case, the posterior is denoted $q_\phi(s_i|\mathbf{y})$.

In both cases, the network is typically trained by minimizing the cross-entropy loss function over the labeled pilot data:

$$\mathcal{L}_{\text{sup}}(\phi) = -\frac{1}{N_p} \sum_{i=1}^{N_p} \log q_\phi(s_i|\mathbf{y}_i \text{ or } \mathbf{y}). \tag{1}$$

Once the parameters $\hat{\phi}$ are learned, the payload data is decoded using the maximum a posteriori rule:

$$\hat{s}_i = \underset{s \in \{1,\dots,K\}}{\operatorname{argmax}} q_{\hat{\phi}}(s \mid \mathbf{y}_i \text{ or } \mathbf{y}), \quad \text{for } i > N_p. \tag{2}$$

The primary limitation of this approach is that it requires a large number of pilots $N_p$ to achieve good performance, which reduces the overall data rate.

## 2.3 THE SEMI-SUPERVISED VARIATIONAL FRAMEWORK

To reduce the dependency on pilots, we adopt a semi-supervised learning (SSL) framework that leverages both the labeled pilot data and the unlabeled payload data. The VAE-based approach in Burshtein & Bery (2023; 2024) provides the foundation for this framework. It involves two parameterized models:

- An **encoder** (or inference model) $q_\phi(s|\mathbf{y})$, which approximates the true posterior $p(s|\mathbf{y})$. This is parameterized by $\phi$.

- A **decoder** (or generative model) $p_\theta(\mathbf{y}|s)$, which models the forward channel process. This is parameterized by $\theta$. See more details regarding the implementation in Appendix C.3.

In the case of channels with memory, the encoder $q_\phi(s_i|\mathbf{y})$ and decoder $p_\theta(\mathbf{y}|\mathbf{s})$ operate on the full sequences $\mathbf{y}$ and $\mathbf{s}$, respectively, to correctly model the temporal dependencies.

The models are trained jointly by minimizing a composite loss function that combines supervised and unsupervised objectives. The full semi-supervised VAE loss function, as derived from Kingma et al. (2014) and applied in Burshtein & Bery (2024), is given by:

$$\mathcal{L}_{\text{SSL-VAE}}(\phi, \theta) = -\frac{\alpha}{N_p}\sum_{i=1}^{N_p}\log q_\phi(s_i|\mathbf{y}_i) - \frac{\gamma}{N_p}\sum_{i=1}^{N_p}\log p_\theta(\mathbf{y}_i|s_i)$$

$$+ \frac{1-\gamma}{N-N_p}\sum_{i=N_p+1}^{N}\left[-\mathbb{E}_{q_\phi(s|\mathbf{y}_i)}[\log p_\theta(\mathbf{y}_i|s)] + D_{KL}(q_\phi(s|\mathbf{y}_i)||p(s))\right], \quad (3)$$

where $\alpha$ and $\gamma$ are hyperparameters that balance the different loss components, and the posterior $q_\phi(s|\mathbf{y})$ represents conditioning on either the individual sample or the entire sequence, $q_\phi(s|\mathbf{y}$ or $\mathbf{y})$. The first two terms represent the supervised losses on the pilot data. The third term is the unsupervised negative Evidence Lower Bound (ELBO) on the payload data, consisting of the reconstruction loss and a regularizing KL divergence term. Minimizing the KL divergence, $D_{KL}(q_\phi(s|\cdot)||p(s))$, encourages the entropy of the encoder's output to be high, preventing it from becoming overconfident on unlabeled data.

Our proposed Constellation-Aware Transformer serves as a direct and powerful replacement for the encoder network $q_\phi(s|\mathbf{y})$ within this exact variational framework. The Transformer is particularly well-suited for channels with memory, as its self-attention mechanism can naturally model the long-range dependencies within the received sequence $\mathbf{y}$ to produce accurate posterior estimates. By providing a richer architectural prior, we aim to learn a much more accurate posterior approximation $q_\phi$, thereby achieving a lower overall loss and superior equalization performance.

## 3 PROPOSED METHOD: THE CONSTELLATION-AWARE TRANSFORMER

Our proposed method is grounded in a Bayesian interpretation of the equalization task, which reveals the necessity of a constellation-aware model. We first present this theoretical motivation and then detail the architecture of our Constellation-Aware Transformer (CAT), a novel deep learning model designed to approximate this ideal Bayesian estimator.

### 3.1 A BAYESIAN VIEW OF CONSTELLATION-AWARE EQUALIZATION

The necessity for a constellation-aware equalizer can be rigorously established within a Hierarchical Bayesian framework. While in any given transmission the constellation is a fixed parameter, a receiver operating under uncertainty can model this lack of knowledge probabilistically. This reframes the problem from simple parameterization to optimal estimation under epistemic uncertainty, revealing that the ideal estimator must inherently process constellation information.

#### 3.1.1 THE HIERARCHICAL GENERATIVE PROCESS

Let us model the complete generative process from the receiver's point of view. Let $\mathfrak{C} = \{\mathcal{C} \subset \mathbb{R}^2 \mid |\mathcal{C}| = K\}$ be the space of all possible constellations. We can model the receiver's beliefs and the physical process as a three-stage hierarchy:

1. **Constellation Prior** $p(\mathcal{C})$: The receiver has a prior belief over the space of constellations, represented by a probability density function (PDF) $p(\mathcal{C})$ where $\mathcal{C} \in \mathfrak{C}$.
2. **Symbol Prior** $p(s|\mathcal{C})$: Once a specific constellation $\mathcal{C}$ is chosen, a symbol $s$ is drawn from it, typically from a uniform distribution over the $K$ points in that constellation.
3. **Channel Likelihood** $p(\mathbf{y}|s)$: The symbol $s$ is transmitted through the channel, resulting in the observation $\mathbf{y}$. $\mathbf{y}$ is conditionally independent of $\mathcal{C}$ given $s$, i.e., $p(\mathbf{y}|s, \mathcal{C}) = p(\mathbf{y}|s)$.

The full joint probability distribution is given by the chain rule: $p(\mathbf{y}, s, \mathcal{C}) = p(\mathbf{y}|s)p(s|\mathcal{C})p(\mathcal{C})$.

### 3.1.2 Deriving the Optimal Estimator Under Uncertainty

The goal is to find the Minimum Mean Squared Error (MMSE) estimator for the symbol $s$ given the observation $\mathbf{y}$, which is the conditional expectation $\mathbb{E}[s|\mathbf{y}]$. To compute this, we must marginalize out the nuisance variable $\mathcal{C}$ by integrating over the entire space of constellations $\mathfrak{C}$. We can express the MMSE estimator using the law of total expectation:

$$\hat{s}_{\text{MMSE}} = \mathbb{E}[s|\mathbf{y}] = \mathbb{E}_{\mathcal{C}|\mathbf{y}}[\mathbb{E}[s|\mathbf{y}, \mathcal{C}]] = \int_{\mathfrak{C}} \mathbb{E}[s|\mathbf{y}, \mathcal{C}]p(\mathcal{C}|\mathbf{y})d\mathcal{C} = \int_{\mathfrak{C}} \left[ \sum_{s \in \mathcal{C}} s \cdot p(s|\mathbf{y}, \mathcal{C}) \right] p(\mathcal{C}|\mathbf{y})d\mathcal{C}. \tag{4}$$

This result demonstrates that the optimal Bayesian estimator must perform two simultaneous inferences: *constellation inference* (computing the posterior PDF $p(\mathcal{C}|\mathbf{y})$) and *symbol conditional estimation* conditioned on a given constellation (the conditional symbol posterior $p(s|\mathbf{y}, \mathcal{C})$).

### 3.1.3 The Special Case: Known Constellation

In our setup, the constellation $\mathcal{C}_{\text{true}}$ is known with certainty. The receiver's prior is therefore a Dirac delta function centered at the true constellation, $p(\mathcal{C}) = \delta(\mathcal{C} - \mathcal{C}_{\text{true}})$. The posterior $p(\mathcal{C}|\mathbf{y})$ is then also a Dirac delta at $\mathcal{C}_{\text{true}}$. The integral in Eq. (4) collapses to evaluating the bracketed term at $\mathcal{C}_{\text{true}}$:

$$\hat{s}_{\text{MMSE}} = \sum_{s \in \mathcal{C}_{\text{true}}} s \cdot p(s|\mathbf{y}, \mathcal{C}_{\text{true}}) = \mathbb{E}[s|\mathbf{y}, \mathcal{C}_{\text{true}}].$$

This is precisely the ideal estimator that a perfect constellation-aware model should target. The purpose of our CAT architecture is to learn an effective approximation of this superior, parameter-aware function.

## 3.2 The Constellation-Aware Transformer (CAT) Architecture

Following recent architectural advances in large language models (LLMs), particularly the recent advancements in efficient Transformer architectures such as Llama 3.1 (Grattafiori et al., 2024), the CAT architecture is composed of a stack of $L$ custom layers, which we term *TransFIRmer blocks*. The architecture is illustrated in Figure 1.

### 3.2.1 Input Representation and Embedding

The input to our model consists of the sequence of $N$ received channel outputs $(\mathbf{y}_1, \ldots, \mathbf{y}_N)$ and the set of $K$ ideal constellation symbols $\{\mathbf{x}(1), \ldots, \mathbf{x}(K)\}$. Both are projected into a processing space of dimension $d_{\text{model}}$ using separate linear embedding layers.

To provide the model with awareness of the sequential nature of the received signals, we inject positional information directly into their representations. Specifically, after the initial linear embedding of the signal sequence, we add fixed sinusoidal positional embeddings, following the original Transformer design (Vaswani et al., 2017). The constellation symbols, which form a set rather than an ordered sequence, are embedded without positional information. The embedded vectors are:

$$\mathbf{z}_i^{\text{sig}} = \text{Embed}_{\text{sig}}(\mathbf{y}_i) + \mathbf{p}_i, i \in \{1, \ldots, N\}, \qquad \mathbf{z}_k^{\text{const}} = \text{Embed}_{\text{const}}(\mathbf{x}(k)), k \in \{1, \ldots, K\},$$

where $\mathbf{p}_i \in \mathbb{R}^{d_{\text{model}}}$ is the sinusoidal positional embedding for the $i$-th position. The full input sequence to the first TransFIRmer block, $\mathbf{Z}^{(0)} \in \mathbb{R}^{(N+K) \times d_{\text{model}}}$, is their concatenation.

### 3.2.2 The TransFIRmer Block

The TransFIRmer block is the core building block of our architecture, modifying the two canonical sub-layers of the Transformer. It sequentially applies: (1) a signal-constellation attention mechanism, and (2) a novel, two-stream feed-forward network inspired by signal processing principles.

**Signal-Constellation Attention Mechanism** The first stage is an efficient multi-query attention (MQA) mechanism (Shazeer, 2019) designed to facilitate deep interaction between the received signals and the ideal constellation symbols. We apply an attention mask $\mathbf{M}$ that configures the signal-to-signal interaction, with modes including 'full' (fully bidirectional), 'causal' (autoregressive), and

'causal channel' (restricting the causal window to the estimated channel length $L$, similarly to sliding window attention in language models (Beltagy et al., 2020)). Although all modes were tested, we found that 'full' attention consistently yielded the best performance and was used for all reported results, likely because it provides a richer context for implicitly estimating the global channel state.

**Two-Stream FIR-Inspired Feed-Forward Network**   The second stage replaces the standard position-wise MLP with a novel two-stream network that processes signal and constellation tokens differently, reflecting their distinct roles.

- **Signal Stream (Bidirectional FIR Filtering):** The sequence of signal token representations, $\mathbf{Z}_{\text{sig}}^{(l)}$, is processed by a pair of 1D convolutional layers acting as Finite Impulse Response (FIR) filters. One learned filter processes the sequence in the forward direction, while a second learned filter processes a time-reversed version of the sequence. The outputs are summed:

$$\text{FFN}_{\text{sig}}\left(\mathbf{Z}_{\text{sig}}\right) = \text{Conv}_{\text{fwd}}\left(\mathbf{Z}_{\text{sig}}\right) + \text{Flip}\left(\text{Conv}_{\text{inv}}\left(\text{Flip}\left(\mathbf{Z}_{\text{sig}}\right)\right)\right). \tag{5}$$

  This structure emulates a non-causal, zero-phase FIR filter, which is the ideal linear processor for deconvolution in block-based communication systems. It provides a powerful and correct inductive bias for learning channel equalization.

- **Constellation Stream (MLP):** In parallel, the set of constellation token representations, $\mathbf{Z}_{\text{const}}^{(l)}$, is processed by a standard two-layer Multi-Layer Perceptron (MLP) for learning complex symbol representations.

The outputs of these two streams are then concatenated and passed through a final residual connection. This two-stream design allows the model to learn an adaptive deconvolution filter for the noisy signals while simultaneously learning feature representations for the clean constellation symbols.

## 3.3 INTEGRATION INTO THE SEMI-SUPERVISED FRAMEWORK

The complete CAT model, comprising the embedding layers and the stack of $L$ TransFIRmer blocks, serves as the encoder network $q_\phi(s|\mathbf{y})$. Its parameters, collectively denoted by $\phi$, are trained end-to-end within the semi-supervised variational framework described in Section 2.3. By replacing a generic encoder with our specialized architecture, we provide the model with strong architectural priors tailored to the equalization task, enabling it to learn a more accurate posterior approximation $q_\phi$ and achieve superior performance with fewer pilot symbols.

## 4 EXPERIMENTS

### 4.1 EXPERIMENTAL SETUP

To evaluate our proposed Constellation-Aware Transformer (CAT), we test it on two categories of channels: a nonlinear memoryless channel with I/Q imbalance and Rayleigh fading, and three standard channels with finite memory (ISI). All experiments use a 16-QAM constellation.

Our primary CAT model consists of a 3-layer stack of TransFIRmer blocks. Inspired by recent trends in optimizing large models, we employ Multi-Query Attention (Shazeer, 2019) and fixed sinusoidal positional embeddings. While other variants such as Multi-Head Attention (Vaswani et al., 2017) or Group-Query Attention (Ainslie et al., 2023), learned embeddings, or RoPE offered similar or marginally better performance in some cases, our chosen configuration significantly reduces the number of model parameters and accelerates both training and inference, making it a more practical choice. Other training hyperparameters, such as learning rate and annealing schedules, are kept consistent with Burshtein & Bery (2024) for a fair comparison, as detailed in Appendix C.

We compare our CAT against a comprehensive set of baselines, largely following the setup in Burshtein & Bery (2024): an Optimal Decoder (ML or BCJR) (Bahl et al., 1974), SSL Monte Carlo EM (MCEM) (Wei & Tanner, 1990), SSL Viterbi EM (Dempster et al., 2018), Simple Decision Directed (SDD), VAE-CNN (Burshtein & Bery, 2024), and the meta-learning algorithm CAVIA (Zintgraf et al., 2019). We also include a Vanilla Transformer baseline, which is adopted for channel equalization (Kunde et al., 2025; Buffelli et al., 2025), to isolate the benefits of our architectural modifications. Brief descriptions of these methods are provided in Appendix D.

To ensure the statistical significance and robustness of our results, each data point presented in our figures and tables represents the mean Symbol Error Rate (SER) averaged over 1000 independent Monte Carlo trials. For each trial, a new channel realization (e.g., new fading coefficients, I/Q imbalance parameters) was randomly generated according to the specified distributions. We computed 95% confidence intervals for all mean SER values and found them to be exceptionally narrow, confirming that the observed performance differences between models are statistically significant and not a result of random fluctuations.

## 4.2 RESULTS ON MEMORYLESS CHANNELS

We first evaluate performance on the nonlinear memoryless channel. All results for our main CAT model use the 'full' attention mask. We found this configuration to be top-performing across both memoryless and memory channels, likely because allowing each signal token to see all other signal tokens provides a richer context for the attention mechanism to implicitly estimate the global channel state before equalization.

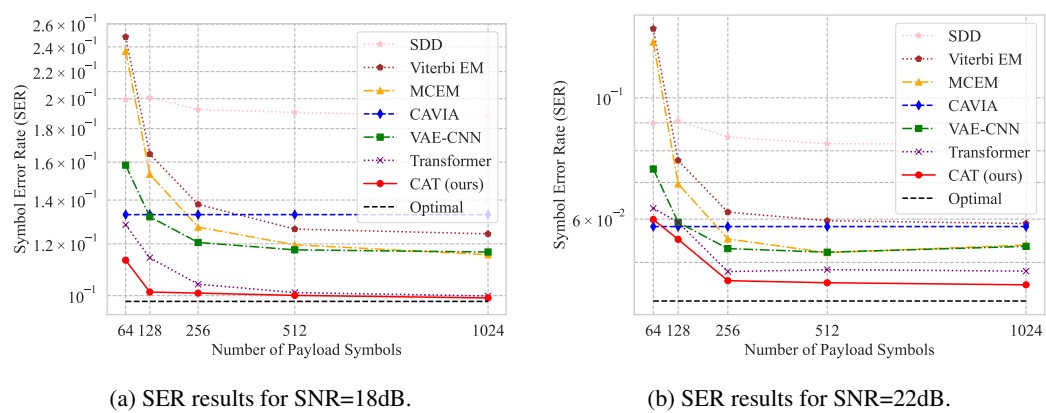

(a) SER results for SNR=18dB.          (b) SER results for SNR=22dB.

Figure 2: Symbol Error Rate (SER) on a memoryless nonlinear channel as a function of the total number of symbols, $N$. The number of pilot symbols is fixed at $N_p = 16$. Our CAT model significantly outperforms all baselines at both SNR=18dB (a) and SNR=22dB (b).

Figure 2 presents the SER as a function of the total number of symbols in a block, $N$, for SNR values of 18dB and 22dB. The results, confirmed to be statistically significant by our rigorous evaluation, clearly demonstrate the superior performance of our proposed CAT model across both SNR regimes. Compared to the VAE-CNN and other baselines, the CAT achieves a significantly lower SER, especially when the total number of symbols is small (e.g., $N = 64$ or $N = 128$). As the number of payload symbols increases, all semi-supervised methods improve, but the CAT consistently maintains a substantial and statistically robust performance gap, closely approaching the Optimal decoder's performance.

## 4.3 RESULTS ON CHANNELS WITH MEMORY (ISI)

To evaluate the CAT's performance on more challenging channels with intersymbol interference (ISI), we adopt three standard channel models taken from Burshtein & Bery (2024):

$$\mathbf{h}_1 = [0.0545 + 0.05j, 0.2832 - 0.11971j, -0.7676 + 0.2788j, -0.0641 - 0.0576j,$$
$$0.0466 - 0.02275j],$$
$$\mathbf{h}_2 = [0.0554 + 0.0165j, -1.3449 - 0.4523j, 1.0067 + 1.1524j,$$
$$0.3476 + 0.3153j],$$
$$\mathbf{h}_3 = [0.0410 + 0.0109j, 0.0495 + 0.0123j, 0.0672 + 0.017j, 0.0919 + 0.0235j,$$
$$0.7920 + 0.1281j, 0.396 + 0.0871j, 0.2715 + 0.048j,$$
$$0.2291 + 0.0415j, 0.1287 + 0.0154j, 0.1032 + 0.0119j].$$

These channels vary in length ($L = 5, 4, 10$ respectively), with a longer impulse response corresponding to more severe ISI and thus a more difficult equalization and channel estimation task. For

Table 1: Symbol Error Rate (SER) on channels with memory ($E_x/N_0 = 17$dB, Payload=256). Our CAT model is compared against the VAE-CNN and a vanilla Transformer. The optimal BCJR performance, which assumes perfect CSI, is included where computationally feasible.

| Channel | Pilots ($N_p$) | VAE-CNN | Transformer | CAT (Ours) | Optimal (BCJR) |
|---|---|---|---|---|---|
| $h^{(1)}$ (L=5) | 16 | **0.2900** | 0.3392 | 0.3580 | |
| | 32 | 0.1251 | 0.1192 | **0.0842** | |
| | 64 | 0.0523 | 0.0610 | **0.0198** | 0.0121 |
| | 128 | 0.0494 | 0.0290 | **0.0156** | |
| $h^{(2)}$ (L=4) | 16 | 0.3447 | 0.3563 | **0.3330** | |
| | 32 | 0.1843 | 0.1593 | **0.1372** | |
| | 64 | 0.1002 | 0.0750 | **0.0340** | 0.0101 |
| | 128 | 0.0869 | 0.0888 | **0.0257** | |
| $h^{(3)}$ (L=10) | 16 | 0.6211 | 0.6481 | **0.6103** | |
| | 32 | 0.3943 | 0.3874 | **0.3426** | |
| | 64 | 0.1709 | 0.1692 | **0.1181** | N/A |
| | 128 | 0.1087 | 0.1022 | **0.0846** | |

this experiment, we use a fixed payload size of 256 symbols and vary the number of pilot symbols $N_p \in \{16, 32, 64, 128\}$. The SNR is set to $E_x/N_0 = 17$dB.

The results are summarized in Table 1. The vanilla Transformer baseline refers to the architecture from Kunde et al. (2025). The results demonstrate a clear trend. While all models struggle in the extremely low-pilot regime ($N_p = 16$), the proposed CAT model begins to significantly outperform both the VAE-CNN and the vanilla Transformer as the number of pilots increases to just 32. For $N_p \geq 64$, the CAT achieves a substantial reduction in SER, often by a factor of 2-3x compared to the next best model. This performance gap is particularly pronounced for the more challenging 10-tap channel ($h_3$), where the CAT's architectural priors provide a distinct advantage.

The table also includes the performance of the optimal BCJR decoder, which assumes perfect and instantaneous channel state information (CSI). As noted in Burshtein & Bery (2024), computing the BCJR performance for the long $h_3$ channel is computationally prohibitive due to the exponential growth of the trellis state space ($16^{L-1}$), hence it is not reported. Our CAT model not only provides the best performance among the learning-based methods but also closes a significant portion of the gap to the theoretical optimal performance, especially with 128 pilots.

## 4.4 ABLATION STUDIES

To empirically validate our design choices, we performed key ablation studies on the memoryless channel (SNR=20dB, $N = 64$). The results (Table 2) confirm the contribution of each component. Our full CAT model achieves the lowest SER of **0.0599**. While differences between top CAT variants are small, their non-overlapping 95% CIs confirm the gains are statistically robust. Removing the inverse FIR or replacing the bidirectional filter with a standard MLP (*CAT with MLP-FFN*) degrades performance, confirming the efficacy of the FIR inductive bias. Crucially, the benefit of constellation awareness is highlighted in two ways: First,

Table 2: Ablation study results (SER at SNR=20dB, $N = 64$). Results are shown with 95% confidence intervals.

| Model Variant | SER ($\pm$ 95% CI) |
|---|---|
| ***Our Full Method*** | |
| CAT (built with TransFIRmer blocks) | **0.0599 $\pm$ 0.0005** |
| *Architecture & Prior Ablations* | |
| CAT without Inverse FIR Filter | 0.0608 $\pm$ 0.0005 |
| CAT with MLP-FFN (No FIR) | 0.0615 $\pm$ 0.0006 |
| Vanilla Transformer (No Prior) | 0.0628 $\pm$ 0.0007 |
| CAT (45° Rotated Prior) | 0.1550 $\pm$ 0.0015 |
| *Attention Mask Ablations* | |
| Self-Only Attention | 0.0621 $\pm$ 0.0007 |
| Causal Attention | 0.0612 $\pm$ 0.0006 |
| *External Baseline* | |
| VAE-CNN (from Burshtein & Bery (2024)) | 0.0741 $\pm$ 0.0009 |

the gap between *CAT with MLP-FFN* and the *Vanilla Transformer* (which lacks the prior) is substantial. Second, providing an incorrect prior (45° rotation) severely degrades performance (SER 0.1550), emphasizing the model's effective utilization of the correct geometric information.

## 5 RELATED WORK

**Semi-Supervised and Unsupervised Channel Equalization.** The challenge of channel equalization with limited pilot data has been extensively studied (Caciularu & Burshtein, 2018; 2020; Lauinger et al., 2022; Song et al., 2023; Nielsen et al., 2025). Recent semi-supervised methods, particularly the VAE-based approach detailed in Zhu et al. (2023); Burshtein & Bery (2023; 2024); Böck et al. (2025), have established a strong baseline by modeling the channel's forward and reverse processes. However, these approaches suffer from two key limitations. Architecturally, the MLP or CNN-based encoders they employ lack the powerful sequence modeling capabilities of modern Transformers (Lu et al., 2022). Conceptually, they must learn the entire problem structure from data alone. From an Information Bottleneck perspective (Tishby et al., 1999), forcing a model to infer the properties of the target symbols—that they belong to a discrete set with a specific, known geometry—is an inefficient use of scarce pilot data. This suggests that a more effective model should not have to *re-discover* this known prior, but rather be explicitly conditioned on it.

**Transformers in Wireless Communications.** The Transformer architecture Vaswani et al. (2017) has recently been explored for various tasks in communications, including channel decoding, estimation, and supervised equalization (Caciularu et al., 2021b; Choukroun & Wolf, 2022; Song et al., 2024; Zhou et al., 2024; Li et al., 2025; Kunde et al., 2025). While demonstrating the power of attention for capturing complex signal dependencies, existing works have two significant gaps. First, they operate almost exclusively in a fully supervised regime, assuming large labeled datasets are available. To our knowledge, the application of Transformers to the more practical semi-supervised equalization setting remains unexplored. Second, these studies typically employ off-the-shelf Transformer architectures. This treats the model as a generic black-box approximator and misses a critical opportunity to incorporate domain knowledge. This approach is analogous to early NLP models that would encode a query and document into separate, fixed-length vectors before comparing them (a "late interaction" model).

A more powerful paradigm, proven successful in diverse and complex NLP tasks, is **early and deep interaction**. Instead of processing a "query" and a "document" in separate streams and only comparing their final, high-level representations (a late-interaction model), recent architectures facilitate fine-grained, token-level attention between them from the very first layer (Conneau & Lample, 2019; Humeau et al., 2019; Gan et al., 2022; Caciularu et al., 2021a; 2023). This principle of avoiding premature summarization is what we are the first to translate to the equalization problem. We append the sequence of received signals as to the ideal constellation symbols, designing an architecture that enables their deep interaction to address the limitations of prior art.

## 6 CONCLUSION

We have presented the Constellation-Aware Transformer (CAT), a novel architecture that achieves state-of-the-art performance in semi-supervised channel equalization by synergistically combining principles from disparate fields: the "early interaction" paradigm from modern NLP, and classical signal processing estimation theory. Its core building block, the TransFIRmer layer, redefines the standard Transformer by integrating two key innovations: (1) a signal-constellation attention mechanism that instantiates the early interaction paradigm by co-processing received signals and ideal constellation symbols from the very first layer, providing an explicit geometric prior throughout the network; and (2) a novel two-stream feed-forward network that applies a specialized bidirectional FIR filter to signal tokens while using a parallel MLP to refine constellation representations.

Our experiments on both nonlinear memoryless and standard ISI channels conclusively demonstrate that this principled design is exceptionally data-efficient. The CAT significantly outperforms VAE, meta-learning, and standard Transformer baselines, particularly in challenging low-pilot regimes where it often reaches near-optimal symbol error rates with as few as 32-64 pilot symbols. The success of this model offers a broader design philosophy for deep learning in the physical sciences: instead of applying generic, black-box architectures, significant gains in performance and data efficiency can be realized by embedding established domain principles directly into the model's structure. This paradigm opens several avenues for future research, including extending the CAT framework to more complex scenarios like MIMO channels and exploring its application to other signal processing tasks.

## ETHICS STATEMENT

We confirm that our work adheres to the code of ethics. This research focuses on foundational improvements in machine learning architectures for the physical layer of communication systems. The study does not involve human subjects, utilize sensitive private data, or employ real-world datasets; all experiments are conducted using standardized, simulated communication channel models. The primary goal of this research is to improve the data efficiency and reliability of communication systems. We anticipate the societal impact to be positive, potentially leading to more robust connectivity and reduced energy consumption in deployed wireless networks by minimizing transmission overhead. While the training of deep learning models requires computational resources, the proposed CAT architecture is relatively lightweight, and its high data efficiency minimizes the computational burden during both training and inference compared to generic architectures. We do not foresee any immediate negative ethical implications stemming directly from this work.

## REPRODUCIBILITY STATEMENT

To ensure the reproducibility of our results, we have provided comprehensive details throughout the paper and the Appendix. The architecture of the proposed Constellation-Aware Transformer (CAT) and the TransFIRmer block is detailed in Section 3. The precise mathematical formulations of the simulated memoryless and ISI channel models used in our experiments are described in Appendix A. Detailed implementation specifics, including model hyperparameters (Appendix C.1), optimizer settings, and the annealing schedules for the semi-supervised learning framework (Appendix C.2), are provided. The architecture of the generative model used within the VAE framework is detailed in Appendix C.3, and descriptions of all baseline methodologies are included in Appendix D. The theoretical analysis supporting the architecture is provided in Appendix E (Theoretical Justification). We utilize standard Symbol Error Rate (SER) metrics, averaged over 1000 Monte Carlo trials as detailed in Section 4.1. To facilitate complete reproduction of our findings, we will make the source code for the CAT model, the simulation environments, and the training scripts publicly available upon acceptance of the paper.

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

## A DETAILED CHANNEL MODELS

### A.1 MEMORYLESS NONLINEAR CHANNEL

The memoryless nonlinear channel model used in our experiments, following Park et al. (2020); Burshtein & Bery (2023; 2024), consists of several stages. First, an ideal transmitted signal $\mathbf{x}_i = (x_i^I, x_i^Q)$ from a 16-QAM constellation is subjected to a nonlinear I/Q imbalance distortion (which mostly stems from hardware imperfections). This creates a distorted signal $\tilde{\mathbf{x}}_i = (\tilde{x}_i^I, \tilde{x}_i^Q)$ according to:

$$\begin{bmatrix} \tilde{x}_i^I \\ \tilde{x}_i^Q \end{bmatrix} = \begin{bmatrix} 1+\epsilon & 0 \\ 0 & 1-\epsilon \end{bmatrix} \begin{bmatrix} \cos\delta & -\sin\delta \\ -\sin\delta & \cos\delta \end{bmatrix} \begin{bmatrix} x_i^I \\ x_i^Q \end{bmatrix}. \tag{6}$$

The imbalance parameters, $\epsilon$ and $\delta$, are constant for each transmission block but are randomly drawn from Beta distributions, specifically $\epsilon = 0.15\epsilon_0$ and $\delta = 15°\delta_0$, where $\epsilon_0, \delta_0 \sim \text{Beta}(5, 2)$.

The resulting complex signal, $\tilde{x}_i^I + j\tilde{x}_i^Q$, is then transmitted over a Rayleigh flat-fading channel. The received complex signal is given by:

$$y_i^I + jy_i^Q = h(\tilde{x}_i^I + j\tilde{x}_i^Q) + n_i, \tag{7}$$

where $h \sim \mathcal{CN}(0, 1)$ is the complex channel gain, which is fixed for the duration of a block, and $n_i \sim \mathcal{CN}(0, \sigma^2)$ is the i.i.d. complex additive white Gaussian noise. The Signal-to-Noise Ratio (SNR) is defined as $10/\sigma^2$, based on the average power of the original 16-QAM constellation. The final received signal used by our models is the real-valued vector $\mathbf{y}_i = (y_i^I, y_i^Q)$.

### A.2 CHANNELS WITH FINITE MEMORY (ISI)

For the experiments involving intersymbol interference, we adopt the channel model from Burshtein & Bery (2023; 2024), which is a two-stage process. First, the ideal signal $\mathbf{x}_i$ undergoes a memoryless nonlinear distortion $g(\cdot)$ to produce $\tilde{\mathbf{x}}_i = g(\mathbf{x}_i)$. For consistency, we use the same I/Q imbalance model described in the previous section for this nonlinearity.

The sequence of distorted signals is then transmitted through a noisy ISI channel. The received signal $\mathbf{y}_i$ is the result of a convolution between the complex channel impulse response $\mathbf{h}$ and the distorted signal sequence, corrupted by additive noise:

$$y_i^I + jy_i^Q = \sum_{l=0}^{L-1} h_l(\tilde{x}_{i-l}^I + j\tilde{x}_{i-l}^Q) + n_i, \tag{8}$$

where $L$ is the length of the channel impulse response and $n_i \sim \mathcal{CN}(0, \sigma^2)$ is complex AWGN. In our simulations, in Section 4.3, the noise variance $\sigma^2$ is set to achieve a target SNR of $E_x/N_0 = 17\text{dB}$.

## B DERIVATION OF THE SEMI-SUPERVISED VAE LOSS

The loss function for the semi-supervised VAE is constructed to leverage both labeled (pilot) and unlabeled (payload) data. The goal is to maximize the log-likelihood of the observed data, which can be expressed as a sum over the labeled and unlabeled sets:

$$\mathcal{L}_{\text{total}} = \sum_{i=1}^{N_p} \log p_\theta(\mathbf{y}_i | s_i) + \sum_{i=N_p+1}^{N} \log p_\theta(\mathbf{y}_i). \tag{9}$$

This is a generative objective. To incorporate the inference network $q_\phi$, we also add a supervised cross-entropy term for the labeled data. The full objective combines these with weighting hyperparameters $\alpha$ and $\gamma$:

$$\mathcal{L}_{\text{full}} = \frac{\alpha}{N_p} \sum_{i=1}^{N_p} \log q_\phi(s_i|\mathbf{y}_i) + \frac{\gamma}{N_p} \sum_{i=1}^{N_p} \log p_\theta(\mathbf{y}_i|s_i) \tag{10}$$

$$+ \frac{1-\gamma}{N - N_p} \sum_{i=N_p+1}^{N} \log p_\theta(\mathbf{y}_i).$$

The final term, $\log p_\theta(\mathbf{y}_i)$ for the unlabeled data, is intractable to compute directly as it requires marginalizing over all possible symbols $s$. We therefore substitute it with its Evidence Lower Bound (ELBO):

$$\log p_\theta(\mathbf{y}_i) \geq \mathbb{E}_{q_\phi(s|\mathbf{y}_i)}[\log p_\theta(\mathbf{y}_i, s) - \log q_\phi(s|\mathbf{y}_i)]. \tag{11}$$

By maximizing this lower bound (equivalent to minimizing its negative), we arrive at the final loss function used for training. After rearranging terms and using the fact that $p_\theta(\mathbf{y}_i, s) = p_\theta(\mathbf{y}_i|s)p(s)$, the negative ELBO becomes:

$$-\text{ELBO} = -\mathbb{E}_{q_\phi(s|\mathbf{y}_i)}[\log p_\theta(\mathbf{y}_i|s)] + D_{KL}(q_\phi(s|\mathbf{y}_i)\|p(s)). \tag{12}$$

Substituting this into the full objective gives the final loss function presented in Eq. (3). For computational tractability, the expectation term is approximated using a single sample from $q_\phi(s|\mathbf{y}_i)$, often implemented with the Gumbel-Softmax reparameterization trick Jang et al. (2017) to maintain differentiability.

## C IMPLEMENTATION AND HYPERPARAMETER DETAILS

### C.1 CAT AND VANILLA TRANSFORMER IMPLEMENTATION

Our Constellation-Aware Transformer (CAT) and the vanilla Transformer baseline share the same core configuration, differing only in their specific architectural components as described in Section 3.

**Architecture.** The models are built with a stack of 3 TransFIRmer (or standard Transformer) layers. The hidden dimension is set to $d_{\text{model}} = 10$, and we use Multi-Query Attention (MQA) (Shazeer, 2019) with a single attention head ($n_{\text{head}} = 1$) for efficiency. For the TransFIRmer layer's two-stream feed-forward network, the bidirectional FIR filter for the signal stream is implemented with two 1D convolutions (one for left-to-right and another for right-to-left convolutions), each using a kernel size of 12. The parallel MLP for the constellation stream uses a single hidden layer that expands the dimension from $d_{\text{model}}$ to $d_{\text{model}}$. Dropout with a rate of $p = 0.1$ is applied within the attention and feed-forward sub-layers. Fixed sinusoidal positional embeddings are used for all experiments.

**Training.** The models are trained using the AdamW optimizer (Loshchilov & Hutter, 2019) with a learning rate of lr $= 10^{-3}$, betas of $(\beta_1, \beta_2) = (0.9, 0.999)$, and a weight decay of 0.01. We employ a linear learning rate scheduler that warms down the learning rate from its initial value to zero over the course of training, which consists of a total of 5000 parameter update steps. The models are trained on mini-batches containing 16 pilot symbols and 32 payload symbols.

### C.2 HYPERPARAMETERS FOR SEMI-SUPERVISED LEARNING

The training of all semi-supervised models (CAT, vanilla Transformer, VAE-CNN, etc.) is governed by the same set of hyperparameters and annealing schedules, ensuring a fair comparison and following the setup in Burshtein & Bery (2023; 2024).

**SSL Loss Weighting.** The composite loss function in Eq. (3) is balanced by two key hyperparameters. The term $\alpha$, which weights the supervised cross-entropy loss on the encoder, is fixed at $\alpha = 0.2$. The term $\gamma$, which balances the supervised reconstruction loss against the unsupervised ELBO, is annealed over the training process. This annealing schedule gradually decreases $\gamma_l$

(where $l$ is the iteration index), shifting the training focus from the reliable pilot data to the more abundant but unlabeled payload data as the model becomes more confident. Specifically, we use $\gamma_l = 1/(1 + \beta_l)$, where $\beta_l = \min(2e^{0.0008(l-1)}, \beta_{\max})$, and $\beta_{\max} = \min((N - N_p)/N_p, 40)$. The value of $\gamma_l$ is updated every 100 iterations.

**Gumbel-Softmax Temperature.** For models utilizing the Gumbel-Softmax reparameterization trick (including our CAT and the VAE-CNN), the temperature $\tau$ is also annealed to transition from a soft, exploratory phase to a hard, decisive phase. The schedule is given by $\tau_l = \max(0.5, e^{-0.001(l-1)})$, with updates occurring every 100 iterations.

### C.3 THE GENERATIVE MODEL ARCHITECTURE

For both our CAT and the vanilla Transformer baseline, we operate within the semi-supervised variational framework, which requires a generative model (or decoder), $p_\theta(\mathbf{y}|s)$, to model the forward channel process. To ensure a fair comparison with prior art, we adopt the generative model architecture directly from the VAE-CNN work in Burshtein & Bery (2023; 2024). The parameters of this model are collectively denoted by $\theta$. The specific architecture differs for memoryless and memory channels.

**Memoryless Channels.** For the memoryless channel, we model $p_\theta(\mathbf{y}_i|s_i)$ as an isotropic Gaussian distribution, $\mathcal{N}(\mathbf{y}_i; \mu_\theta(\mathbf{x}(s_i)), \sigma_\theta^2(\mathbf{x}(s_i))\mathbf{I})$. The mean $\mu_\theta$ and log-variance $\log \sigma_\theta^2$ are produced by a decoder network. This network is a Multi-Layer Perceptron (MLP) which takes the ideal constellation signal $\mathbf{x}(s_i) \in \mathbb{R}^2$ as input, passes it through a series of hidden layers with ReLU activations, and finally uses two separate linear heads to output the 2-dimensional mean and log-variance vectors.

**Channels with Memory (ISI).** For channels with memory, the generative model is designed to explicitly capture the two-stage process of a transmitter nonlinearity followed by a linear ISI channel. The model first applies a memoryless nonlinear function $g(\cdot)$, parameterized by a small MLP, to each ideal symbol $\mathbf{x}_i$ in the input sequence $\mathbf{s}$ to produce a sequence of distorted signals $\tilde{\mathbf{x}}$. This sequence is then convolved with a learnable Finite Impulse Response (FIR) filter, which models the complex channel impulse response $\mathbf{h}$. The real and imaginary parts of the filter taps are stored as two separate learnable parameter vectors. The output of this convolution provides the mean of the Gaussian distribution for the received sequence. The noise is modeled as i.i.d. Gaussian with a single learnable variance parameter $\sigma^2$. Thus, the parameters $\theta$ for the generative model in the ISI case consist of the weights of the MLP $g$, the channel filter taps, and the scalar noise variance.

## D BASELINE METHODOLOGIES

### D.1 SIMPLE DECISION DIRECTED (SDD)

The SDD algorithm is a classical two-stage semi-supervised method (Burshtein & Bery, 2023; 2024).

1. **Initial Training:** A standard neural network decoder, $q_\phi(s|\mathbf{y})$, is first trained exclusively on the labeled pilot data $\{(\mathbf{y}_i, s_i)\}_{i=1}^{N_p}$ by minimizing the cross-entropy loss from Eq. (1). Let the resulting parameters be $\hat{\phi}_0$.

2. **Pseudo-Labeling and Retraining:** The trained model is used to generate "hard" pseudo-labels for the unlabeled payload data: $\hat{s}_i = \arg\max_s q_{\hat{\phi}_0}(s|\mathbf{y}_i)$ for $i > N_p$. The model's parameters are then fine-tuned by training on a combined dataset of original pilots and pseudo-labeled payload data, minimizing a weighted cross-entropy loss.

### D.2 VITERBI EM

The Viterbi EM algorithm (Dempster et al., 2018) is a hard-decision variant of the Expectation-Maximization (EM) algorithm, as described in Burshtein & Bery (2024). It uses a generative model of the channel, $p_\theta(\mathbf{y}|s)$, parameterized by $\theta$, and iterates between two steps:

1. **E-Step (Expectation):** Given the current estimate of the generative model's parameters $\theta^{(t-1)}$, generate hard decisions (pseudo-labels) for the payload data by choosing the most likely symbol according to the current model: $\hat{s}_i^{(t)} = \mathrm{argmax}_s\, p_{\theta^{(t-1)}}(\mathbf{y}_i|s)$.

2. **M-Step (Maximization):** Update the generative model's parameters by minimizing the reconstruction loss (negative log-likelihood) over a combined dataset of the original pilots and the newly generated pseudo-labels from the E-step, yielding $\theta^{(t)}$.

This process is repeated for a fixed number of iterations, gradually refining the channel model.

## D.3 VAE-SSL (VAE-CNN)

This is the state-of-the-art semi-supervised method proposed in Burshtein & Bery (2023; 2024), which we refer to as VAE-CNN based on its typical implementation. It is a variational autoencoder-based framework that jointly trains two models:

- An **encoder** $q_\phi(s|\mathbf{y})$, which acts as the primary decoder. For channels with memory, this is typically implemented with a Convolutional Neural Network (CNN).

- A **decoder** $p_\theta(\mathbf{y}|s)$, which is a generative model that learns the forward channel process.

The two networks are trained simultaneously using a composite semi-supervised loss function (detailed in Eq. (3) that combines a supervised objective on the pilot data with an unsupervised, Evidence Lower Bound (ELBO) objective on the payload data. This allows the model to leverage the entire data block to learn a robust representation of the channel.

## D.4 CAVIA META-LEARNING

Fast Context Adaptation via Meta-Learning (CAVIA) (Zintgraf et al., 2019) is a meta-learning algorithm designed for rapid adaptation to new tasks. In our context, each channel realization is a "task".

- **Meta-Training:** The model is trained on data from a large number of previous channel blocks $\{(\mathbf{y}^{(m)}, s^{(m)})\}_{m=1}^M$. The goal is to learn a set of shared parameters $\phi$ that are common across all channels, while a small, task-specific "context vector" $\mathbf{z}^{(m)}$ is learned for each individual channel.

- **Meta-Testing (Adaptation):** When a new channel block arrives, the shared parameters $\phi$ are frozen. The model then rapidly infers a new context vector $\mathbf{z}_{\mathrm{new}}$ by training only on the few available pilot symbols from the new block.

- **Decoding:** The final decoder uses both the shared parameters $\phi$ and the adapted context vector $\mathbf{z}_{\mathrm{new}}$ to decode the payload data of the new block.

CAVIA's strength lies in its ability to learn a good "general" model that can be quickly specialized, making it highly effective in few-shot (low pilot) scenarios, provided that past channel data is available.

# E  THEORETICAL JUSTIFICATION: OPTIMAL ESTIMATION AND ARCHITECTURAL ALIGNMENT

The advantage of the Constellation-Aware Transformer (CAT) architecture can be rigorously justified from three complementary perspectives: a Hierarchical Bayesian framework, statistical learning theory (hypothesis spaces), and the functional decomposition of the optimal equalizer. We establish the necessity of accurate constellation information for optimal estimation and demonstrate that the CAT architecture is structurally aligned with theoretically optimal filtering. We note that similar, but more general conclusions (not restricted to specific architectures) were shown by Böck et al. (2024); Böck et al. (2024).

### E.1 Bayesian Estimation and the Cost of Constellation Uncertainty

We further analyze the Hierarchical Bayesian framework introduced in Section 3.1. This framework allows us to quantify the impact of uncertainty about the constellation on the Minimum Mean Squared Error (MMSE) estimation.

#### E.1.1 The Optimal Estimator and Posterior Dependence

Recall from Section 3.1 that the optimal MMSE estimator under uncertainty is derived by marginalizing out the constellation $\mathcal{C}$ using the law of total expectation (Eq. (4)):

$$\hat{s}_{\text{MMSE}}(\mathbf{y}) = \mathbb{E}[s|\mathbf{y}] = \mathbb{E}_{\mathcal{C} \sim p(\mathcal{C}|\mathbf{y})}[\mathbb{E}[s|\mathbf{y}, \mathcal{C}]]. \tag{13}$$

This formulation demonstrates that the optimal estimator depends critically on the true constellation posterior $p(\mathcal{C}|\mathbf{y})$.

#### E.1.2 The Impact of Approximate Constellation Knowledge

In practical systems, or when using domain-agnostic models that do not have explicit access to the constellation, the exact posterior $p(\mathcal{C} \mid \mathbf{y})$ might be unavailable. A model might instead rely on an *implicit approximation* of this posterior information, learned from data. We can view the implemented estimator as replacing the exact marginalization in Eq. (13) by a mixture built from some approximation $r(\mathcal{C} \mid \mathbf{y})$ that the model realizes:

$$\hat{s}_r(\mathbf{y}) = \mathbb{E}_{\mathcal{C} \sim r(\mathcal{C}|\mathbf{y})}[\mathbb{E}[s \mid \mathbf{y}, \mathcal{C}]]. \tag{14}$$

The following lemma quantifies the suboptimality introduced by this approximation.

**Lemma 1** (Estimator-Gap Bound). *Assume symbols $s$, considering a finite-energy constellation, lie in a bounded set with $\|s\|_2 \leq S_{\max}$. Let $p(\mathcal{C} \mid \mathbf{y})$ be the true constellation posterior and let $r(\mathcal{C} \mid \mathbf{y})$ be an approximation. Define $\hat{s}_{\text{MMSE}}(\mathbf{y})$ and $\hat{s}_r(\mathbf{y})$ as above. Then, for every observation $\mathbf{y}$,*

$$\|\hat{s}_r(\mathbf{y}) - \hat{s}_{\text{MMSE}}(\mathbf{y})\|_2^2 \leq 2 S_{\max}^2 \, \text{KL}\big(r(\mathcal{C} \mid \mathbf{y}) \,\|\, p(\mathcal{C} \mid \mathbf{y})\big). \tag{15}$$

*Proof.* We aim to bound the squared L2 distance between the optimal MMSE estimator, $\hat{s}_{\text{MMSE}}(\mathbf{y})$, utilizing the true posterior $p(\mathcal{C}|\mathbf{y})$, and the approximate estimator, $\hat{s}_r(\mathbf{y})$, utilizing the approximation $r(\mathcal{C}|\mathbf{y})$. We denote these distributions as $p$ and $r$ for brevity.

Let $g(\mathcal{C})$ be the conditional MMSE estimator given a specific constellation $\mathcal{C}$:

$$g(\mathcal{C}) := \mathbb{E}[s \mid \mathbf{y}, \mathcal{C}].$$

The estimators can be expressed as expectations of $g(\mathcal{C})$:

$$\hat{s}_{\text{MMSE}}(\mathbf{y}) = \int g(\mathcal{C})p(d\mathcal{C}),$$

$$\hat{s}_r(\mathbf{y}) = \int g(\mathcal{C})r(d\mathcal{C}).$$

We first establish that $g(\mathcal{C})$ is bounded. Since the L2 norm $\|\cdot\|_2$ is a convex function, we can apply Jensen's inequality:

$$\|g(\mathcal{C})\|_2 = \|\mathbb{E}[s \mid \mathbf{y}, \mathcal{C}]\|_2 \leq \mathbb{E}[\|s\|_2 \mid \mathbf{y}, \mathcal{C}].$$

Given the assumption that $\|s\|_2 \leq S_{\max}$ almost surely, the expectation is also bounded by $S_{\max}$. Thus, $\|g(\mathcal{C})\|_2 \leq S_{\max}$ for all $\mathcal{C}$.

We now analyze the L2 norm of the difference between the two estimators. By the linearity of integration, we can combine them into a single integral over the signed measure $(r - p)$:

$$\|\hat{s}_r(\mathbf{y}) - \hat{s}_{\text{MMSE}}(\mathbf{y})\|_2 = \left\|\int g(\mathcal{C})r(d\mathcal{C}) - \int g(\mathcal{C})p(d\mathcal{C})\right\|_2$$

$$= \left\|\int g(\mathcal{C})(r - p)(d\mathcal{C})\right\|_2.$$

Next, we apply the generalized triangle inequality for integrals (which is a form of Jensen's inequality for the norm function), stating that $\|\int f\,d\mu\| \le \int \|f\|\,d|\mu|$, where $|\mu|$ is the total variation measure of the signed measure $\mu$.

$$\left\|\int g(\mathcal{C})(r-p)(d\mathcal{C})\right\|_2 \le \int \|g(\mathcal{C})\|_2\,|r-p|(d\mathcal{C}).$$

We now utilize the established bound $\|g(\mathcal{C})\|_2 \le S_{\max}$:

$$\int \|g(\mathcal{C})\|_2\,|r-p|(d\mathcal{C}) \le \int S_{\max}\,|r-p|(d\mathcal{C})$$

$$= S_{\max}\int |r-p|(d\mathcal{C}).$$

The term $\int |r-p|(d\mathcal{C})$ is the L1 distance between the probability measures $r$ and $p$. This is related to the Total Variation (TV) distance, defined as $\|r-p\|_{\mathrm{TV}} = \frac{1}{2}\int |r-p|(d\mathcal{C})$. Substituting this definition:

$$S_{\max}\int |r-p|(d\mathcal{C}) = 2S_{\max}\|r-p\|_{\mathrm{TV}}.$$

Thus far, we have shown $\|\hat{s}_r(\mathbf{y}) - \hat{s}_{\mathrm{MMSE}}(\mathbf{y})\|_2 \le 2S_{\max}\|r-p\|_{\mathrm{TV}}$. To relate the TV distance to the KL divergence, we invoke Pinsker's inequality (Pinsker, 1964), which states that $\|r-p\|_{\mathrm{TV}} \le \sqrt{\frac{1}{2}\mathrm{KL}(r\|p)}$.

$$\|\hat{s}_r(\mathbf{y}) - \hat{s}_{\mathrm{MMSE}}(\mathbf{y})\|_2 \le 2S_{\max}\sqrt{\frac{1}{2}\mathrm{KL}(r\|p)}.$$

Finally, squaring both sides yields the stated bound:

$$\|\hat{s}_r(\mathbf{y}) - \hat{s}_{\mathrm{MMSE}}(\mathbf{y})\|_2^2 \le \left(2S_{\max}\sqrt{\frac{1}{2}\mathrm{KL}(r\|p)}\right)^2$$

$$= 4S_{\max}^2 \cdot \frac{1}{2}\mathrm{KL}(r\|p)$$

$$= 2S_{\max}^2\,\mathrm{KL}(r\|p).$$

$\square$

### E.1.3 Implications for Constellation-Aware Design

Lemma 1 provides a strong motivation for the CAT architecture. As discussed in Section 3.1.3, in our setup the constellation $\mathcal{C}_{\mathrm{true}}$ is known. The true posterior is therefore a Dirac delta function, $p(\mathcal{C}|\mathbf{y}) = \delta(\mathcal{C} - \mathcal{C}_{\mathrm{true}})$.

A perfect constellation-aware model like CAT explicitly utilizes this knowledge, effectively trying to set its approximation $r(\mathcal{C}|\mathbf{y}) = p(\mathcal{C}|\mathbf{y})$. The KL divergence is zero, and the estimator gap vanishes. The resulting estimator is the ideal $\mathbb{E}[s|\mathbf{y}, \mathcal{C}_{\mathrm{true}}]$.

Conversely, a domain-agnostic model (like a Vanilla Transformer) must infer the constellation from scarce pilot data. It learns an implicit approximation $r(\mathcal{C}|\mathbf{y})$. Lemma 1 shows that the performance of such a model is fundamentally limited by its ability to accurately estimate the true constellation structure. The inefficiency of using data to learn a known prior results in a non-zero KL divergence and thus a performance gap.

### E.2 The Superiority of Constellation-Aware Hypothesis Spaces

We now analyze the problem through the lens of Minimum Mean Squared Error (MMSE) estimation. The objective is to estimate the true symbol $s$, drawn from constellation $\mathcal{C}$, from an observation

$\mathbf{y} \in \mathbb{C}^N$. The goal is to minimize the MSE, $\mathbb{E}\left[\|s - \hat{s}\|^2\right]$. The optimal estimator is the conditional expectation:

$$\hat{s}_{\text{MMSE}} = \mathbb{E}[s|\mathbf{y}].$$

By Bayes' rule, $p(s|\mathbf{y}) \propto p(\mathbf{y}|s)p(s)$. The prior $p(s)$ is defined by the constellation $\mathcal{C}$. Thus, the optimal estimator is intrinsically dependent on $\mathcal{C}$. When approximating $\mathbb{E}[s|\mathbf{y}]$ using deep learning, we choose between different hypothesis spaces.

- **Standard (Domain-Agnostic) Approach:** The estimator $\hat{s} = g(\mathbf{y})$ is chosen from a class of functions $\mathcal{G}_{\text{std}}$ that map the observation space to the symbol space without explicit knowledge of the constellation structure.
- **Constellation-Aware (Parametric) Approach:** The estimator $\hat{s} = \psi(\mathbf{y}, \mathcal{C})$ is chosen from an expanded class $\mathcal{G}_{\text{CA}}$ that explicitly accepts the constellation $\mathcal{C}$ as a parameter.

**Theorem 1** (Advantage of the Parametric Hypothesis Space). *Let $s$ be a symbol drawn from a finite constellation $\mathcal{C}$ and observed as $\mathbf{y}$. Define the hypothesis classes:*

$$\mathcal{G}_{std} = \{g : \mathbb{C}^N \to \mathbb{C}\}, \quad \mathcal{G}_{CA} = \{\psi : \mathbb{C}^N \times \mathcal{P}(\mathbb{C}) \to \mathbb{C}\}.$$

*where $\mathcal{P}(\mathbb{C})$ is the space of possible constellations. The achievable MMSE satisfies:*

$$MMSE_{CA} = \inf_{\psi \in \mathcal{G}_{CA}} \mathbb{E}\left[\|s - \psi(\mathbf{y}, \mathcal{C})\|^2\right] \leq \inf_{g \in \mathcal{G}_{std}} \mathbb{E}\left[\|s - g(\mathbf{y})\|^2\right] = MMSE_{std}.$$

*Proof.* The class of standard estimators $\mathcal{G}_{\text{std}}$ is a subset of the constellation-aware estimators $\mathcal{G}_{\text{CA}}$. For any function $g \in \mathcal{G}_{\text{std}}$, one can define a function $\psi \in \mathcal{G}_{\text{CA}}$ as $\psi(\mathbf{y}, \mathcal{C}) = g(\mathbf{y})$ for all $\mathcal{C}$. This function $\psi$ ignores its second argument. Thus, $\mathcal{G}_{\text{std}} \subseteq \mathcal{G}_{\text{CA}}$. Since the infimum of a function over a superset cannot be larger than the infimum over a subset, it follows directly that $\text{MMSE}_{\text{CA}} \leq \text{MMSE}_{\text{std}}$. $\square$

A strict improvement ($\text{MMSE}_{\text{CA}} < \text{MMSE}_{\text{std}}$) is realized if and only if the optimal estimator $\psi^*$ is a function of $\mathcal{C}$. This is true when $\mathcal{C}$ is the true constellation, as the ideal estimator $\mathbb{E}[s|\mathbf{y}]$ depends fundamentally on the prior $p(s)$ defined by $\mathcal{C}$. Conversely, if an irrelevant parameter is provided, the MMSE remains the same. Theorem 1 formally establishes that providing the known constellation as input grants access to a superior solution space.

This theoretical advantage is strongly validated by the empirical ablation studies presented in Section 4.4 (Table 2). The Vanilla Transformer, representing an estimator from $\mathcal{G}_{\text{std}}$, must implicitly learn the constellation geometry from scarce pilot data, resulting in a significantly higher Symbol Error Rate (SER) compared to the CAT architecture, which leverages $\mathcal{G}_{\text{CA}}$. Furthermore, the superiority formalized here relies critically on the accuracy of the provided prior $\mathcal{C}$. As demonstrated in the ablations, when CAT is supplied with an incorrect prior (a $45°$ rotated constellation), its performance degrades severely, falling below even the agnostic baseline. This confirms that the model is indeed utilizing the provided geometric information effectively, and that the realization of the theoretical gains depends on the fidelity of the injected domain knowledge.

### E.2.1 Architectural Alignment with Optimal Wiener Filtering

We provide a rigorous justification for the Constellation-Aware Transformer (CAT) architecture by demonstrating its structural alignment with optimal equalization and detection strategies. We show that CAT is designed to realize the optimal MIMO Wiener Filter (Wiener & Hopf, 1931) for channel inversion and the optimal Matched Filter for detection.

**1. The Optimal Receiver Structure.** The channel model, including I/Q distortion (represented by a $2 \times 2$ matrix $\mathbf{G}$) and ISI (with complex taps $\{h_l\}$), results in an effective $2 \times 2$ real-valued MIMO channel impulse response, $\mathbf{H}_l = \mathbf{M}(h_l)\mathbf{G}$, where $\mathbf{M}(h_l)$ is the real matrix equivalent of the complex tap $h_l$. The overall transmission for a block of $N$ symbols can be expressed in matrix form as $\mathbf{Y} = \mathcal{H}\mathbf{X} + \mathbf{N}$, where $\mathcal{H}$ is the $2N \times 2N$ block Toeplitz convolution matrix constructed from the taps $\{\mathbf{H}_l\}$.

*(i) Optimal Equalization (The MIMO Wiener Filter):* The Linear Minimum Mean Square Error (LMMSE) equalizer for this system is the MIMO Wiener Filter (WF). Its coefficients are given by the

Wiener-Hopf equations (Wiener & Hopf, 1931; Lawrie & Abrahams, 2007). Here, $E_s = \mathbb{E}[\|\mathbf{x}_i\|^2]$ represents the average energy per transmitted symbol, calculated as $E_s = \frac{1}{K}\sum_{k=1}^{K}\|\mathbf{x}(k)\|^2$ for a uniform symbol distribution. The filter is defined as:

$$\mathbf{W}_{\text{WF}}^T = E_s \mathcal{H}^T (E_s \mathcal{H}\mathcal{H}^T + \sigma^2 \mathbf{I}_{2N})^{-1}. \tag{16}$$

This filter performs several critical functions: **Temporal Deconvolution** (inverting ISI), **Spatial Alignment** (correcting I/Q imbalance and channel rotations), and **Noise Optimization**. It requires estimating the channel's autocorrelation matrix $\mathbf{R}_{YY} \propto E_s \mathcal{H}\mathcal{H}^T + \sigma^2 \mathbf{I}$ and applying its inverse, a process known as whitening or decorrelation.

*(ii) Optimal Detection (The Matched Filter):* After optimal equalization, the resulting estimate $\hat{\mathbf{x}}_i \in \mathbb{R}^2$ is best detected using the Matched Filter. This process correlates the estimate with each possible ideal constellation vector $\mathbf{x}(k) \in \mathbb{R}^2$ and selects the one with the highest correlation, adjusted by an energy bias term:

$$\text{Decision} = \underset{k}{\arg\max} \left( \hat{\mathbf{x}}_i^T \mathbf{x}(k) - \frac{1}{2}\|\mathbf{x}(k)\|^2 \right). \tag{17}$$

**Proposition 1** (Structural Realization of the Optimal Receiver by CAT). *The Constellation-Aware Transformer (CAT) architecture possesses the necessary inductive biases and structural components to efficiently learn the MIMO Wiener Filter (or its generalization for unknown I/Q distortion) for equalization and explicitly implement the Matched Filter bank for detection.*

*Proof.* We demonstrate how the components of CAT realize these optimal strategies.

1. **Learning the MIMO Wiener Filter: The TransFIRmer Block Synergy.** The equalization stage ($\mathcal{T}_\phi$) in CAT aims to learn the channel inversion corresponding to $\mathbf{W}_{\text{WF}}$. This is achieved through the synergy of the FIR-FFN and the Self-Attention mechanism.

   *a. The MIMO FIR Inductive Bias (FIR-FFN):* The Wiener Filter $\mathbf{W}_{\text{WF}}$ is fundamentally a MIMO FIR filter. The TransFIRmer block's FIR-FFN (Eq. (5)) provides a learnable, bidirectional FIR structure. This is precisely the functional class required to implement the temporal deconvolution and spatial alignment (I/Q correction) of the MIMO WF, ensuring the learned transformation is physically plausible.

   *b. Dynamic Statistics Estimation and Whitening (Self-Attention):* The Wiener solution requires estimating the channel statistics ($\mathbf{R}_{YY}$) and applying its inverse for whitening. The self-attention mechanism is uniquely suited for this dynamic estimation, as it computes pairwise interactions:

   $$\text{Attention Scores} \propto (\mathbf{Z}W_Q)(\mathbf{Z}W_K)^T.$$

   This operation is structurally analogous to computing an empirical estimate of the local covariance structure $\hat{\mathbf{R}}_{YY}$, capturing how the channel correlates the input sequence. While the Transformer does not explicitly compute the matrix inverse $\mathbf{R}_{YY}^{-1}$, the attention mechanism *functionally realizes* the whitening transformation. Driven by the MSE loss minimization, the network parameters adapt to decorrelate the input features, as this is the optimal strategy for equalization.

   *c. Adaptive Conditional Wiener Filtering:* The components synergize within the TransFIRmer block. Self-attention dynamically estimates the local channel statistics and conditions the features (implicit whitening). The FIR-FFN then utilizes these conditioned features to apply the precise, structured filtering. By integrating these, the TransFIRmer block effectively learns a *conditional* Wiener filter, adapted dynamically to the specific channel realization within the block.

   *d. Adaptation to Unknown I/Q Distortion:* The channel includes I/Q imbalance, modeled as an unknown linear transformation $\mathbf{G}$ (Appendix A) that varies per block. The equalizer must rapidly adapt to this unknown $\mathbf{G}$, which is embedded within the overall channel matrix $\mathcal{H}$. The stacked architecture of CAT excels at this, learning a complex mapping from the observed signal statistics (estimated by attention and dependent on $\mathbf{G}$) to the optimal filter coefficients needed to invert both I/Q distortion and ISI.

2. **Implementing the Matched Filter: Constellation-Aware Attention.** The detection stage ($g_\theta$) implements the optimal decision rule (Eq. (17)).

   *a. The Inefficiency of Implicit Learning:* Domain-agnostic architectures (e.g., Vanilla Transformers) must use scarce pilot data to learn the constellation points $\mathbf{x}(k)$ in their final classifier weights. This is inefficient, as the constellation geometry $\mathcal{C}$ is known.

   *b. Explicit Implementation and Decoupling:* The Constellation-Aware Attention mechanism explicitly calculates the correlation between the equalized signal features (Queries, $\mathbf{z}_i^{(L)}$) and the constellation representations (Keys, $\mathbf{c}_k^{(L)}$):

$$\alpha_{i,k} = \frac{(W_Q \mathbf{z}_i^{(L)})^T (W_K \mathbf{c}_k^{(L)})}{\sqrt{d_{\text{model}}}} \tag{18}$$

   The architecture is designed to align embeddings with physical quantities ($\mathbf{z}_i^{(L)} \to \hat{\mathbf{x}}_i$ and $\mathbf{c}_k^{(L)} \to \mathbf{x}(k)$). Through appropriate parameterization, the attention score directly computes the Matched Filter's correlation term, $\hat{\mathbf{x}}_i^T \mathbf{x}(k)$. By providing $\mathcal{C}$ as input, CAT structurally implements the optimal detector. This crucially **decouples** the detection task from the equalization task, allowing the network's entire learning capacity to focus on the complex challenge of approximating the adaptive MIMO Wiener filter.

   $\square$

### E.3 CONCLUSION: SYNERGISTIC ALIGNMENT AND REDUCED SAMPLE COMPLEXITY

The rigorous justification for the superior performance of the CAT architecture stems from the synergistic alignment of its components with optimal communication theory, supported by the Bayesian analysis and learning theory frameworks.

Lemma 1 quantifies the critical need for accurate constellation information, showing that estimation error is bounded by the divergence between the model's implicit belief and the true constellation posterior. Theorem 1 establishes that providing this information explicitly grants access to a superior hypothesis space. Proposition 1 demonstrates that CAT is structurally designed to leverage this advantage by realizing the fundamental components of an optimal MIMO receiver:

1. The FIR-FFN provides the exact MIMO FIR structure required for the Wiener filter, enabling both **temporal deconvolution** and **spatial alignment**.

2. Self-attention dynamically estimates the necessary channel statistics and implicitly performs adaptive decorrelation (whitening).

3. The Constellation-Aware Attention mechanism explicitly implements the Matched Filter bank by utilizing the known constellation geometry $\mathcal{C}$.

This alignment provides a powerful, physically grounded inductive bias that significantly reduces the complexity of the learning task. Domain-agnostic models must use scarce pilot data to rediscover both the optimal equalization strategy and the detection geometry.

In contrast, CAT structurally embeds the optimal detection strategy. This crucially **decouples** detection from equalization, allowing the network's entire learning capacity to focus solely on the complex task of equalization—adapting the filter parameters to the specific, unknown channel realization. By constraining the search space to functions that are theoretically optimal, CAT reduces estimation variance and achieves significantly lower sample complexity, explaining the substantial empirical gains observed.

