# OpenReview forum: "A Constellation-Aware Transformer for Nonlinear Channel Equalization"
_ICLR.cc/2026/Conference — ICLR 2026 Conference Withdrawn Submission_

### Official Review · Reviewer_vShY · 2025-10-23

**Soundness:** 3
**Presentation:** 3
**Contribution:** 2
**Rating:** 4
**Confidence:** 3

**Summary:**

The manuscript describes a new transformer based neural architecture for non-linear channel equalisation.

**Strengths:**

In theoretical point of view, the study is interesting. Formulation seems sound and their extension of architecture seems sound  and results are also ok.

**Weaknesses:**

The thing which puzzles me quite a lot and is that what is actually the intended use case for the approach they consider? Based on description in section 2.3, a neural network is trained based on pilots and payload data in semi-supervised manner. If I interpret this correctly, this would mean that a transformed needs to be trained or fine-tuned every time when channel conditions change. If you consider common wireless communication frameworks, such as 5G or WiFi, this happens all the time especially if mobile device or terminal is moving. In those system, you would need to process a large numbers of symbols in milliseconds and the complexity to train continuously a transformer model would be out of limits of any practical implementation with several orders of magnitude. This comment of course also applies Burstein & Bery's work, which is the basis of this work, but authors could comment this also here for the audience of ICLR.

Minor specific comments:
- PAs can also have memory effects (e.g GaN RF amplifiers suffers from it)
- Figure 1 font size could be increased.

**Questions:**

- As mentioned above, authors could clarify possible use cases for their methods,
- Continuing from that, would there be a way to separate sources of nonlinear effects such as PA and the wireless channel (which is usually linear) in a way that transformed would be just trained one for each transmitter with a nonlinear PA?  This would perhaps make more sense in practical point of view.

---

### Official Review · Reviewer_kxLq · 2025-10-24

**Soundness:** 3
**Presentation:** 3
**Contribution:** 3
**Rating:** 4
**Confidence:** 4

**Summary:**

This paper proposes a modification to the transformer architecture to incorporate domain knowledge regarding the constellation structure and channel effects into the design choices.

**Strengths:**

1. Using cross-attention between the received vector and the constellation early on to provide more context to the decoder is a clever and interesting approach.

2. The FIR-inspired design approach is interesting. Replacing the FFN block with a convolutional filter-like block is appropriate for the problem under consideration.

3. Empirical results are convincing and tested well on both memoryless and ISI channels.

4. The ablation studies reasonably back up the claims presented.

**Weaknesses:**

1. Relatively simple idea with overall contributions more suitable for a communication venue. Particularly because of the empirical validation on experiments, which are very specific to wireless communication.

2. Given the broad adoption and availability of realistic channel simulators such as Sionna, experimenting with such channel simulators instead of limiting to synthetic channels would be more convincing.

3. Interesting and well-executed, but the work lacks sufficient core ML innovation to meet a machine-learning venue’s bar.

**Questions:**

1. Can you perform some experiments with a more realistic channel, such as TDL/CDL channels with mobility, in Sionna? Given the problem of poor equalization in OFDM systems during mobility, and a well-established line of work on "neural receivers" and "neural precoding", discussion and differentiation with these works would be appreciated.

2. Instead of considering the uncoded transmission, can the BER/BLER be measured in the presence of channel coding? Quantification of gains on coded transmission is more relevant and might give a better picture of the contribution of each component, given that the current performance of the model for different variants is very similar.

---

### Official Review · Reviewer_bxnZ · 2025-10-28

**Soundness:** 3
**Presentation:** 3
**Contribution:** 2
**Rating:** 2
**Confidence:** 4

**Summary:**

This paper introduces the Constellation-Aware Transformer (CAT). Each TransFIRmer module is an independent processing unit that redesigns the standard Transformer encoder by integrating two key innovations, both derived from the following principles:1. Constellation-Aware Attention (CAT) 2. Feedforward networks inspired by finite impulse response. Experiments demonstrate that it achieves state-of-the-art performance in channel decoding.

**Strengths:**

This paper is well-written, with a clear and logical structure. The comparisons presented in the experimental section are also very clear, and the appendix provides a substantial amount of background information.

**Weaknesses:**

The greatest contribution of this paper lies in the innovative design of the network, but I believe that the novelty of the proposed design is insufficient, and the explanation for why such a design was chosen is also inadequate. Firstly, the Signal-Constellation Attention Mechanism seems to demonstrate through experiments that full attention is better than causal attention. However, the use of uncertain wording like "likely because" suggests that it might be due to the ability of full attention to capture global information. Whether there is any correlation between information across different channels, or whether local designs (e.g., Swin Transformer) might be better than global designs, is not addressed in this paper. Secondly, in the FFN, the authors simulate Finite Impulse Response filters using 1D convolution. This does not bring any prior knowledge. In my opinion, the improved performance gain is due to the combination of convolutional and linear layers, which has already been proven feasible in other transformer design papers.

The comparison methods used in the experimental phase are also insufficient. In the related work section, the paper claims that current transformers used in the wireless field are all traditional transformers with little design innovation. However, this is not the case. For example, the paper "Joint Channel Estimation and Feedback with Masked Token Transformers in Massive MIMO Systems" employs a channel selection method. Another paper, "Transformer-Empowered CSI Feedback for Massive MIMO Systems," also uses transformers for channel estimation. In the experimental phase, this paper only compares against ordinary transformers, indicating that the experiments are very preliminary.

**Questions:**

see above

---

### Official Review · Reviewer_vF6C · 2025-11-06

**Soundness:** 2
**Presentation:** 2
**Contribution:** 2
**Rating:** 2
**Confidence:** 4

**Summary:**

- The paper introduces a novel transformer-based architecture designed for channel equalization.
- The contribution is twofold: 1) The receiver employs a transformer network that utilizes signal constellation properties to enhance decoding performance. 2) The paper proposes a novel, FIR-inspired feed-forward network within the transformer layers to mimic channel deconvolution.

**Strengths:**

- The core concepts presented are both interesting and novel.
- The inclusion of experiments on channels with memory is a valuable addition to the evaluation.

**Weaknesses:**

- The experimental results are limited to high SNR values (17, 18, 20, and 22 dB). The lack of experiments across a wider SNR range makes it difficult to assess performance under varied channel conditions.
- The evaluation on channels with memory is restricted to only three specific channel models.
- There is no comparison or normalization based on computational complexity or parameter count. It is unclear if the proposed CAT transformer outperforms the VAE-CNN baseline due to architectural superiority or simply having more parameters.
- For both memoryless and memory channels, the CAT model appears to be trained and tested on the same channel realization. This approach raises concerns about the system's generalizability, as real-world channel conditions are dynamic rather than static.

**Questions:**

- The selection of specific channels with memory is attributed to Bushtein & Bery. Is there any specific reasoning behind choosing these particular channels for experimentation?

---

### Note · Authors · 2025-11-30

I have read and agree with the venue's withdrawal policy on behalf of myself and my co-authors.